# Mapping global dynamics of benchmark creation and saturation in artificial intelligence

Simon Ott[1,5], Adriano Barbosa-Silva[1,2,5], Kathrin Blagec[1], Jan Brauner [3,4] & Matthias Samwald [1] ✉

Benchmarks are crucial to measuring and steering progress in artificial intelligence (AI). However, recent studies raised concerns over the state of AI benchmarking, reporting issues such as benchmark overfitting, benchmark saturation and increasing centralization of benchmark dataset creation. To facilitate monitoring of the health of the AI benchmarking ecosystem, we introduce methodologies for creating condensed maps of the global dynamics of benchmark creation and saturation. We curate data for 3765 benchmarks covering the entire domains of computer vision and natural language processing, and show that a large fraction of benchmarks quickly trends towards near-saturation, that many benchmarks fail to find widespread utilization, and that benchmark performance gains for different AI tasks are prone to unforeseen bursts. We analyze attributes associated with benchmark popularity, and conclude that future benchmarks should emphasize versatility, breadth and real-world utility.

Benchmarks have become crucial to the development of artificial intelligence (AI). Benchmarks typically contain one or more datasets and metrics for measuring performance. They exemplify and—explicitly or implicitly—define machine learning tasks and goals that models need to achieve. Models achieving new state-of-the-art (SOTA) results on established benchmarks receive widespread recognition. Thus, benchmarks do not only measure, but also *steer* progress in AI.

Looking at individual benchmarks, one can identify several phenomenologically different SOTA dynamics patterns, such as continuous growth, saturation/stagnation, or stagnation followed by a burst (Fig. 1).

The continuous growth pattern is marked by a steady increase of the SOTA curve over the years. The saturation/stagnation pattern is characterized by initial growth followed by a long-term halt in improvement of the SOTA. This may either be caused by a lack of improvement in technical capability (technological stagnation), a lack of research interest in the benchmark (research intensity stagnation),

or by an inability to further improve on the benchmark because its inherent ceiling has already been reached (saturation). The stagnation followed by burst pattern is marked by a flat or only slightly increasing SOTA curve, eventually followed by a sharp increase. This might indicate a late breakthrough in tackling a certain type of task.

In recent years, a sizable portion of novel benchmarks in key domains such as NLP quickly trended towards saturation[1]. Benchmarks that are nearing or have reached saturation are problematic, since either they cannot be used for measuring and steering progress any longer, or—perhaps even more problematic—they see continued use but become misleading measures: actual progress of model capabilities is not properly reflected, statistical significance of differences in model performance is more difficult to achieve, and remaining progress becomes increasingly driven by over-optimization for specific benchmark characteristics that are not generalizable to other data distributions[2,3]. Hence, novel benchmarks need to be created to complement or replace older benchmarks.

[1]Institute of Artificial Intelligence, Medical University of Vienna. Währingerstraße 25a, 1090 Vienna, Austria. [2]ITTM S.A.—Information Technology for Translational Medicine, Esch-sur-Alzette 4354, Luxembourg. [3]Oxford Applied and Theoretical Machine Learning (OATML) Group, Department of Computer Science, University of Oxford, Oxford, UK. [4]Future of Humanity Institute, University of Oxford, Oxford, UK. [5]These authors contributed equally: Simon Ott, Adriano Barbosa-Silva. ✉e-mail: matthias.samwald@meduniwien.ac.at

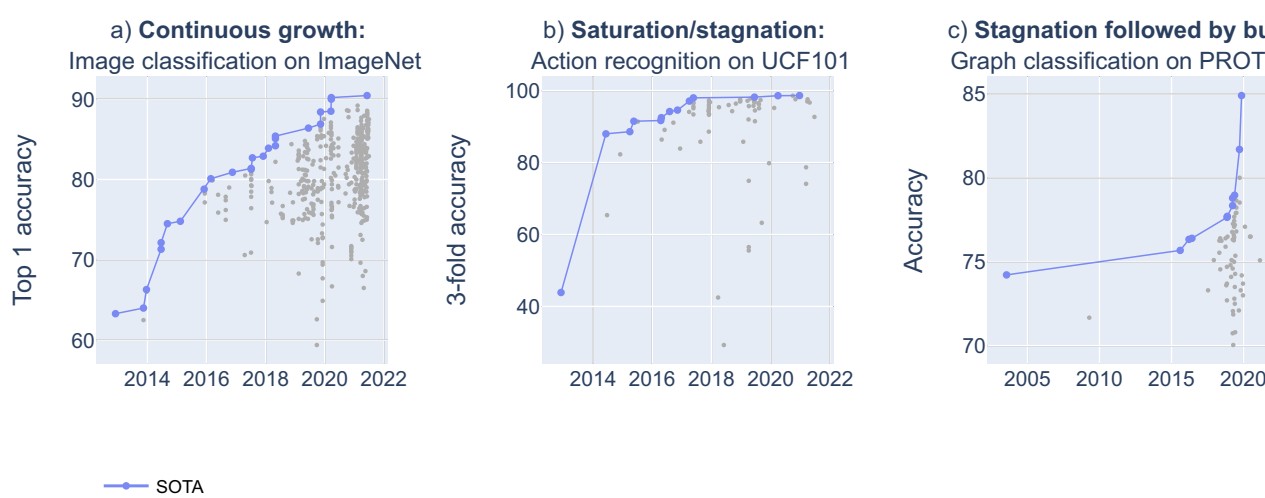

**Fig. 1 | Examples of within-benchmark dynamics patterns. a** Continuous growth (ImageNet benchmark[5]), **b** saturation/stagnation (UCF101 benchmark[24]), **c** stagnation followed by burst (PROTEINS benchmark[25]). The line shows the trajectory of SOTA results, dots show all benchmarks results (including those not setting new SOTA).

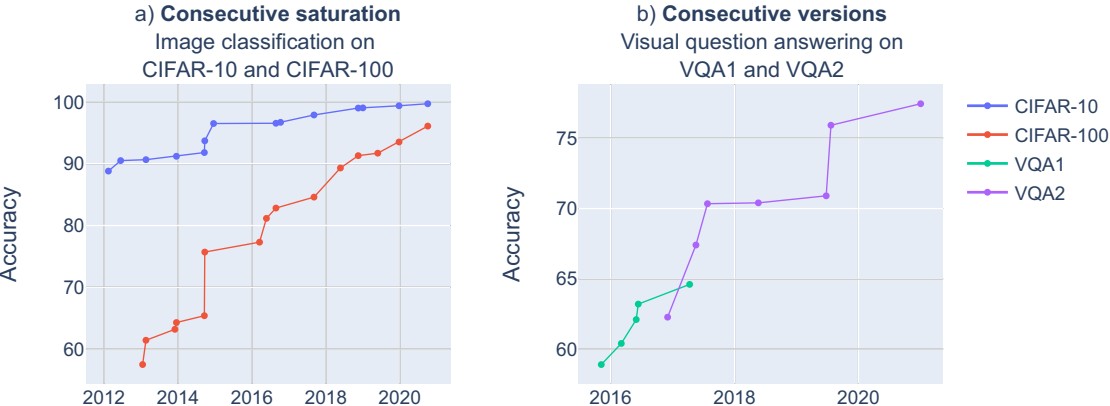

**Fig. 2 | Across-benchmark dynamics patterns. a** Consecutive saturation (CIFAR-10 vs. CIFAR-100;[6] note that CIFAR-100 has not fully reached saturation yet), **b** Consecutive versions (VQA 1 vs. VQA 2[26]).

These phenomena generate patterns across two or more related benchmarks over time, such as clear-cut consecutive versions or consecutive saturation of benchmarks (Fig. 2).

Some recent work analyzed the global evolution of AI capabilities as measured through benchmarks. The annual AI index[4] investigated progress in performance exemplified through selected benchmarks per task type (e.g., ImageNet[5] for Image Classification or SQuAD for natural language understanding) and compared to human performance. As an example, in the AI index report for 2021, it noted that gains in computer vision benchmarks were flattening, while natural language processing (NLP) models were outpacing available benchmarks for question answering and natural language understanding.

Martínez-Plumed et al. analyzed the community dynamics behind 25 popular AI benchmarks, such as CIFAR-100[6] and SQuAD1.1[7,8]. They found that the 'SOTA front' and SOTA jumps were dominated by long-term collaborative hybrid communities, formed predominantly by American or Asian universities together with tech giants, such as Google or Facebook.

Koch et al. analyzed trends in dataset use and repurposing across a large number of AI benchmarking efforts[9]. They discovered that a large fraction of widely used datasets were introduced by only a few high-profile organizations, that this disparity increased over time and that some of these datasets were increasingly re-purposed for novel tasks. However, they also found that NLP was an exception to this trend, with greater than average introduction and use of novel, task-specific benchmarks.

There still remain substantial gaps in our understanding of the global dynamics of benchmarking efforts. How well are different classes of AI tasks represented through benchmarking efforts? How are benchmarks created, used and abandoned over time? How quickly do benchmarks become saturated or stagnant, thereby failing to capture or guide further progress? Can trends across AI tasks and application domains be identified? Why are some benchmarks highly utilized while others are neglected?

In this work, we investigate these questions and expand on previous work by exploring methods for mapping the dynamics of benchmark creation, utilization and saturation across a vast number of AI tasks and application domains. We extract data from Papers With Code (paperswithcode.com), the largest centralized repository of AI benchmark results, and conduct extensive manual curation of AI task type hierarchies and performance metrics. Based on these data, we analyze benchmark dynamics of two highly productive AI domains of recent years: computer vision and NLP.

## Results

We included 3765 benchmarks across 947 distinct AI tasks in our analysis. We found that for a significant fraction of the benchmarks in our dataset, only few results were reported at different time points in

different studies (Table 1). For example, 1318 NLP benchmarks have at least one result reported, but only 661 (50%) of these have results reported at three or more different time points.

## SOTA curve diversity and dynamics
In order to explore the diversity of real-world within-benchmark SOTA dynamics in a data-driven way, we used Self Organizing Maps (SOM)—a type of Artificial Neural Network able to convert complex, nonlinear statistical relationships between high-dimensional data items into simple geometric relationships on a low-dimensional display—to cluster individual metrics curves based on their shapes. Only SOTA trajectories with at least five entries over at least one year were considered.

Figure 3 displays the three clusters discovered for benchmarks in computer vision and NLP for all metrics. In total, 1079 metric trajectories of 654 benchmarks were assigned to one of three clusters. Cluster 1 (460 trajectories) most closely resembles the

phenomenology of continuous growth. Cluster 2 (378 benchmarks), corresponds to the saturation/stagnation scenario. In this cluster, values close to the ceiling of all results are observed very soon in the time series and limited remaining growth in performance is recorded afterwards. Finally, cluster 3 (241 benchmarks) most closely resembles the stagnation followed by breakthrough scenario.

We analyzed the number of benchmarks reporting new SOTA results vs. active benchmarks reporting any results over time for NLP (Fig. 4) and computer vision (Suppl. Fig. 3). For both NLP and computer vision, the number of benchmarks in the dataset started to rise in 2013, with a notable acceleration of growth in benchmarks reporting SOTA results in 2017-2018 and a slowdown of growth after 2018. There is a strong stagnation of the number of active and of SOTA-reporting benchmarks in 2020, which is more marked for NLP. The peak numbers of active benchmarks in the dataset were highest in 2020 (432 for NLP, 1100 for computer vision), demonstrating that the availability of benchmarks for computer vision remained significantly higher compared to NLP.

To understand in greater detail how benchmark creation and saturation unfold across the great variety of tasks that are addressed by global AI research, we devised methodologies for normalizing and visualizing benchmark dynamics, as described below.

## Creating global maps of AI benchmark dynamics
Comparing SOTA trajectories across a wide variety of tasks, benchmarks and performance metrics is not trivial: How significant is a certain increment in performance? What constitutes a markedly unimpressive result (i.e., the 'floor' of our expectations)? What would constitute the best result realistically possible (i.e., the 'ceiling' of our expectations)?

**Table 1 | Descriptive statistics of reported results over time for specific benchmarks and AI tasks**

|  | NLP | Computer vision | Total |
|---|---|---|---|
| Benchmarks with ≥1 reported result | 1318 | 2447 | 3765 |
| Benchmarks with ≥3 results at different time points (% of above) | 661 (50%) | 1274 (52%) | 1935 (51%) |
| AI tasks with ≥1 reported result | 346 | 601 | 947 |
| AI tasks with ≥3 results at different time points (% of above) | 197 (57%) | 386 (64%) | 583 (62%) |

A single task can be represented through several benchmarks.
*NLP* natural language processing.

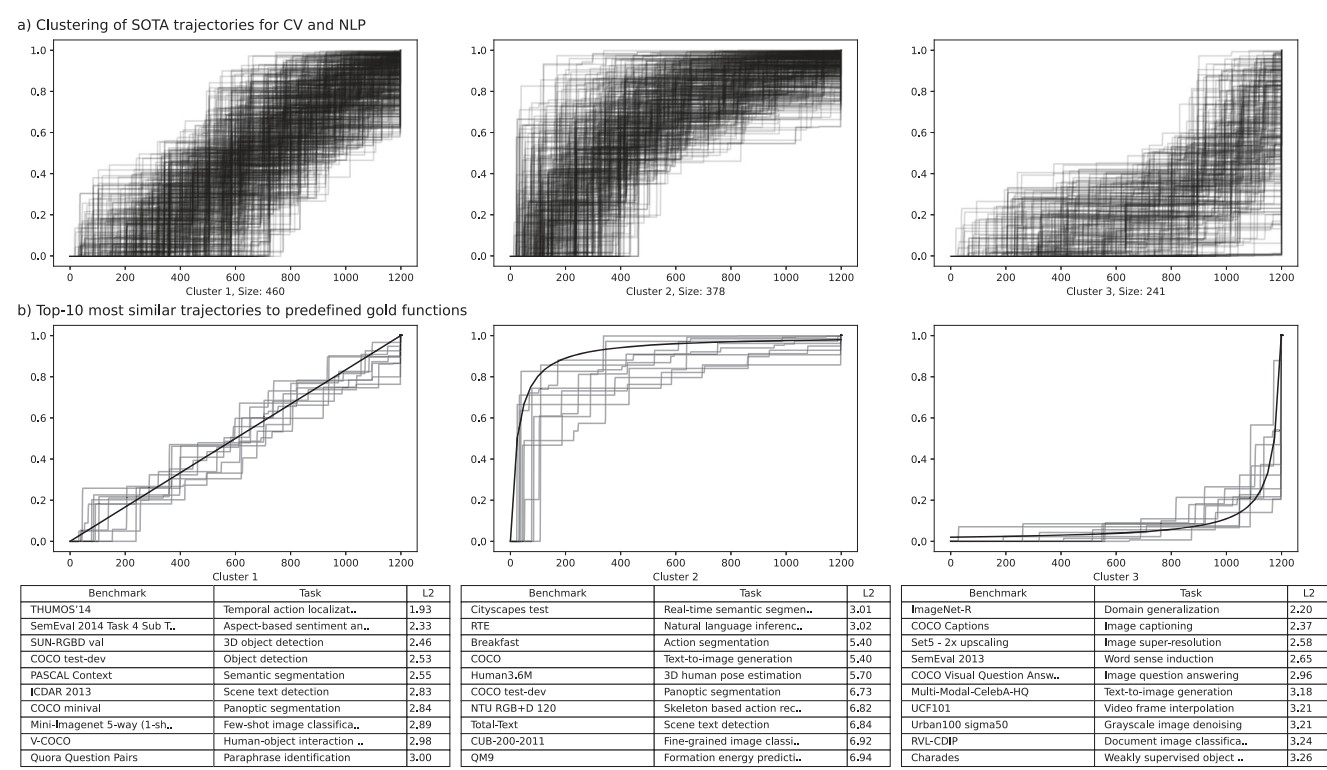

a) Clustering of SOTA trajectories for CV and NLP

Cluster 1, Size: 460 — Cluster 2, Size: 378 — Cluster 3, Size: 241

b) Top-10 most similar trajectories to predefined gold functions

Cluster 1 — Cluster 2 — Cluster 3

| Benchmark | Task | L2 |
|---|---|---|
| THUMOS'14 | Temporal action localizat.. | 1.93 |
| SemEval 2014 Task 4 Sub T.. | Aspect-based sentiment an.. | 2.33 |
| SUN-RGBD val | 3D object detection | 2.46 |
| COCO test-dev | Object detection | 2.53 |
| PASCAL Context | Semantic segmentation | 2.55 |
| ICDAR 2013 | Scene text detection | 2.83 |
| COCO minival | Panoptic segmentation | 2.84 |
| Mini-Imagenet 5-way (1-sh.. | Few-shot image classifica.. | 2.89 |
| V-COCO | Human-object interaction .. | 2.98 |
| Quora Question Pairs | Paraphrase identification | 3.00 |

| Benchmark | Task | L2 |
|---|---|---|
| Cityscapes test | Real-time semantic segmen.. | 3.01 |
| RTE | Natural language inferenc.. | 3.02 |
| Breakfast | Action segmentation | 5.40 |
| COCO | Text-to-image generation | 5.40 |
| Human3.6M | 3D human pose estimation | 5.70 |
| COCO test-dev | Panoptic segmentation | 6.73 |
| NTU RGB+D 120 | Skeleton based action rec.. | 6.82 |
| Total-Text | Scene text detection | 6.84 |
| CUB-200-2011 | Fine-grained image classi.. | 6.92 |
| QM9 | Formation energy predicti.. | 6.94 |

| Benchmark | Task | L2 |
|---|---|---|
| ImageNet-R | Domain generalization | 2.20 |
| COCO Captions | Image captioning | 2.37 |
| Set5 - 2x upscaling | Image super-resolution | 2.58 |
| SemEval 2013 | Word sense induction | 2.65 |
| COCO Visual Question Answ.. | Image question answering | 2.96 |
| Multi-Modal-CelebA-HQ | Text-to-image generation | 3.18 |
| UCF101 | Video frame interpolation | 3.21 |
| Urban100 sigma50 | Grayscale image denoising | 3.21 |
| RVL-CDIP | Document image classifica.. | 3.24 |
| Charades | Weakly supervised object .. | 3.26 |

**Fig. 3 | Empirical data on diversity of within-benchmark dynamics. a** Diversity of within-benchmark dynamics patterns observed for the metrics across all benchmarks from NLP and computer vision. For assignments of individual benchmarks among the clusters see Supplementary Data 5). **b** Top-10 most similar trajectories to predefined gold trajectories representing linear growth ($f(x) = x$ where

$\{x \in \mathbb{N} 1 \le x \le 50\}$), early saturation ($f(x) = -1/x$ where $\{x \in \mathbb{N} 1 \le x \le 50\}$) and stagnation followed by breakthrough ($f(x) = -1/x$ where $\{x \in \mathbb{N} -50 \le x \le -1\}$). As similarity metric we use the euclidean distance between trajectory and gold function.

Natural language processing

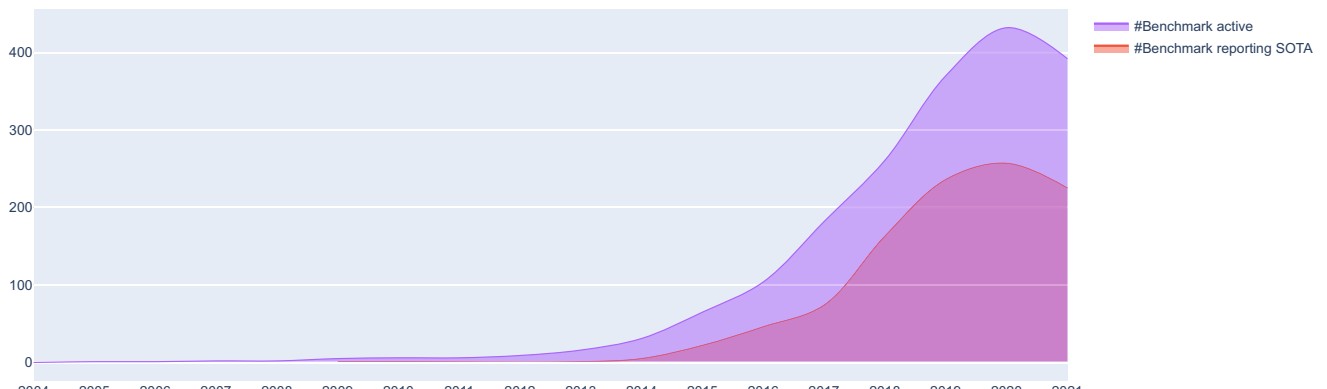

**Fig. 4 | Development of benchmark activity in NLP over time.** Shown are the number of active benchmarks (i.e., benchmarks for which any novel results were reported) vs. number of benchmarks reporting novel state-of-the-art (SOTA) results over time for NLP tasks. A similar plot for computer vision is available in the associated online material and supplementary material.

Different performance metrics can inherently cover widely different value ranges. For example, for the performance metric accuracy, the inherent lowest value would be 0%, while the inherent highest value would be 100%. However, this is often less helpful for judging benchmark results than one might hope: For a balanced dataset with two classes, the floor should rather be set to 50%, as this would be the accuracy achieved by a random classifier. For an unbalanced dataset, the inherent floor might be another value—i.e., it would be highly dataset specific. Similar concerns can also be raised about the potential ceiling value: for example, a perfect accuracy score of 100% might never be achievable even by the best hypothetical model because of limitations inherent in the dataset (e.g., mislabeled examples in the test set).

Arguably, the best solution for judging and comparing trajectories would be an in-depth manual analysis and curation of all benchmarks, where floor values are determined by trivially simple prediction algorithms and ceiling values are determined through gold standards (e.g., expert human curators). Unfortunately, such curated data are not available for the vast majority of benchmarks. Moreover, even purported gold standard test sets can have severe limitations. For example, many recent NLP benchmarks were quickly saturated with some systems reaching above-human performance on test sets[1]—but further analyses reveal that models achieving very high performance often did so through recognizing benchmark-inherent artifacts that did not transfer to other data distributions[1,3].

To easily scale our analysis to all benchmarks without requiring cost-prohibitive per-benchmark curation of performance metrics value ranges, we normalized and calibrated SOTA benchmark results in a data-driven way. As the basis of further analyses, we calculated the relative improvement (i.e., increase in performance) for individual metrics (e.g., accuracy). We achieved this by comparing the stepwise increment from the first (A, anchor) to the last reported result (M, maximum) in each step of a SOTA trajectory.

We define relative improvement (r) as:

$$r_i = \frac{R_i - R_{i-1}}{M - A}, i>1 \tag{1}$$

where relative improvement ($r_i$) is the ratio of the difference between the current result ($R_i$) minus the previous result ($R_{i-1}$) over the difference between the last result (M, maximum) minus the first result (A, anchor). Because we need the anchor value ($i = 1$) as reference for the $r_i$ calculation of the subsequent values, we only consider the calculation

of $r$ from the second value ($i = 2$) in the trajectory onwards. Figure 5 exemplifies this calculation and visualizes the resulting values for SOTA accuracy results of five AI models reported from October 2015 until April 2017 for the visual question answering benchmark 'VQA v1 test-dev'.

The methodology exemplified in Fig. 5 was applied to all AI tasks in NLP and computer vision to create global SOTA trajectory maps (generated data are available in Supplementary Data 3 and 4). To condense the visual representation, data items for the same task and month were aggregated by selecting the maximum value.

Figure 6 displays the global SOTA trajectory map for NLP. Here, every dash represents an anchor, i.e., the first result of a newly established benchmark. The subsequent icons depict the relative improvements for different benchmarks belonging to each task. We grouped tasks based on their superclasses extracted from the ontology structure we created during data curation (see Methods section), placing related tasks adjacent to each other. For example, "Semantic analysis" is the superclass of "Semantic textual similarity" and "Word sense disambiguation". A similar global SOTA trajectory map for computer vision is available in Supplementary Fig. 1.

Interactive versions of these plots that allow for displaying details for each data item can be accessed online through a webpage (https://openbiolink.github.io/ITOExplorer/) and Jupyter notebooks (Code 2, Code availability).

In NLP the tasks of information extraction, sentiment analysis, language modeling and question answering had significant density of novel SOTA results the earliest (2014–2016). It is noteworthy that none of the tasks completely ceased to produce SOTA activity once they became established. Relative SOTA improvements were modest until 2018. There was a slight clustering of large relative SOTA improvements around 2018–2019—a possible interpretation being that this was when AI language capabilities experienced a boost while benchmarks were not yet saturated.

In computer vision, high research intensity and continuous progress on image classification benchmarking (Supplementary Fig. 1) started in 2013. This is earlier than most other AI tasks, as those were the first application areas in which deep learning started to excel. Notable later advances happened in 3D vision processing (since 2016), image generation (since 2017) and few-shot learning (2018–2019). In terms of relative SOTA improvements, the map for CV shows a wide array of patterns in benchmark dynamics across different AI tasks that elude simple narratives about benchmark intensity and progress.

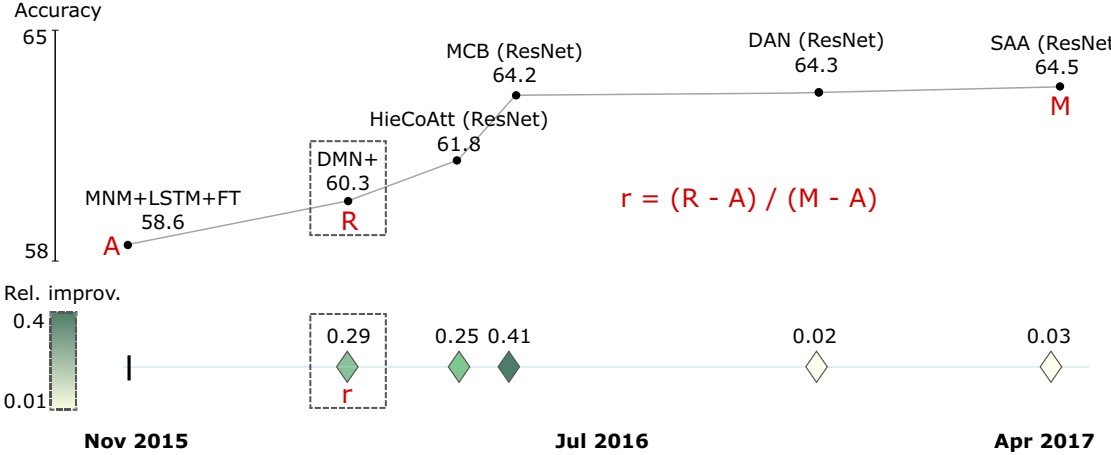

**Fig. 5 | Example of calculating the relative improvement in SOTA for the 'VQA v1 test-dev' benchmark.** Top: The SOTA curve displays accuracy results achieved by different models over time. Bottom: The values of the SOTA curve rendered as relative improvement $r$, calculated as the ratio of the obtained result ($R$) minus the previous result over the difference between the final ($M$, maximum) and first ($A$, anchor) accuracy values. The first result ($A$) is displayed as a vertical dash in the trajectory, whereas the remaining SOTA jumps are depicted as icons with color corresponding to relative improvement.

To further visualize the dynamics of benchmark creation, progression, saturation/stagnation and eventual abandonment, we devised a global AI benchmark lifecycle map, exemplified in Fig. 7 for the NLP domain. The lifecycle map classifies each benchmark into one of four classes every year: (a) New benchmark: benchmark reporting its first result this years, (b) Benchmark reporting SOTA: established benchmark that reports at least one SOTA result, (c) Benchmark reporting no SOTA/no results: established benchmarks that does not report any results, or does report results but none of them establish a new SOTA, and (d) Disbanded benchmark: a benchmark that does not report further results from a given year onwards. In the lifecycle map, every class is represented as an icon, while the size of the icon represents the number of benchmarks falling into this category. Each benchmark can only fall into a single category for each year.

The figure and a related figure for computer vision are also available as interactive graphs on the web (https://openbiolink.github.io/ITOExplorer/).

The benchmark lifecycle map for NLP (Fig. 7) shows that a few benchmarks across most tasks were established early (before 2015), but only a small number of novel SOTA results were reported for these benchmarks during this period. Establishment of novel benchmarks strongly accelerated in 2015–2018, and was most marked in question answering, information extraction, text classification and text summarization. The years 2018 and 2019 saw the establishment of many novel benchmarks for a wide variety of further tasks, as well as the reporting of SOTA results for large numbers of benchmarks. Establishment of novel benchmarks was reduced in 2020, and concentrated on high-level tasks associated with inference and reasoning, likely because of increasing model capabilities in these areas. From 2019, no novel SOTA results (or no results at all) were reported for a large number of benchmarks, and this phenomenon was not particularly biased towards any specific types of tasks.

The lifecycle map for computer vision (Supplementary Fig. 2) shows a first wave of benchmark establishment for the tasks of image clustering and image classification around 2013, followed by several other tasks in 2014. It is noteworthy that even tasks established early—such as image classification and semantic segmentation—demonstrated high benchmark activity and novel SOTA results well into 2021, and especially for image classification this was accompanied by an ongoing establishment of novel few-shot benchmarks. Tasks for most other benchmarks were established in 2015–2019.

For both NLP and computer vision, the number of distinct benchmarks strongly differs between tasks. Only a very fraction of benchmarks was disbanded in the years up to 2020. A larger number of benchmarks was reported as disbanded from 2020 (i.e., have no reported results in 2020 or after). The number of benchmarks classified as disbanded is highest in 2021, but this is likely partially influenced by the cutoff date of the dataset used in the analysis (mid 2022).

## Dataset popularity is distributed very unevenly

We selected all datasets used to benchmark NLP or computer vision tasks and which had first reported results in the Papers With Code dataset in 2018. We analyzed the distribution of dataset popularity, measured by the number of scientific papers utilizing each dataset for NLP. We found distributions to be heavy-tailed, i.e., a small set of benchmark datasets was used to generate a large number of benchmark results, as demonstrated in Fig. 8 for NLP datasets. The top 22% of NLP datasets and top 21% of computer vision datasets were utilized by the same number of papers as the remaining datasets for each domain. The disparity becomes even greater when analyzing all datasets in Papers With Code, regardless of their first recorded entry: Here, the top 10% of NLP and top 5% of computer vision datasets were utilized by the same number of papers as the remaining datasets.

## Quantifying Papers With Code dataset completeness

While Papers With Code is the largest dataset of AI benchmark results by a wide margin, it cannot provide a full coverage of all existing AI benchmarks. We conducted a small-scale study to estimate the completeness of the Papers With Code dataset regarding SOTA result trajectories.

We randomly sampled 10 benchmark datasets from NLP and 10 benchmark datasets from computer vision in the dataset, resulting in a total of 20 randomly sampled datasets (listed in Supplementary Data 7). Querying Google Scholar, we found that the total size of the combined corpus of papers introducing the datasets and all their citing papers was 7595. Out of the citing papers, we randomly sampled 365 papers (sample size chosen to yield a margin of error of 5% in the analysis).

We inspected and annotated these 365 papers to determine whether each paper contained results on the benchmark of the cited dataset paper. If this was the case, we compared the reported result with the Papers With Code dataset to determine if the paper reported a result that was SOTA at the time and was not currently covered by

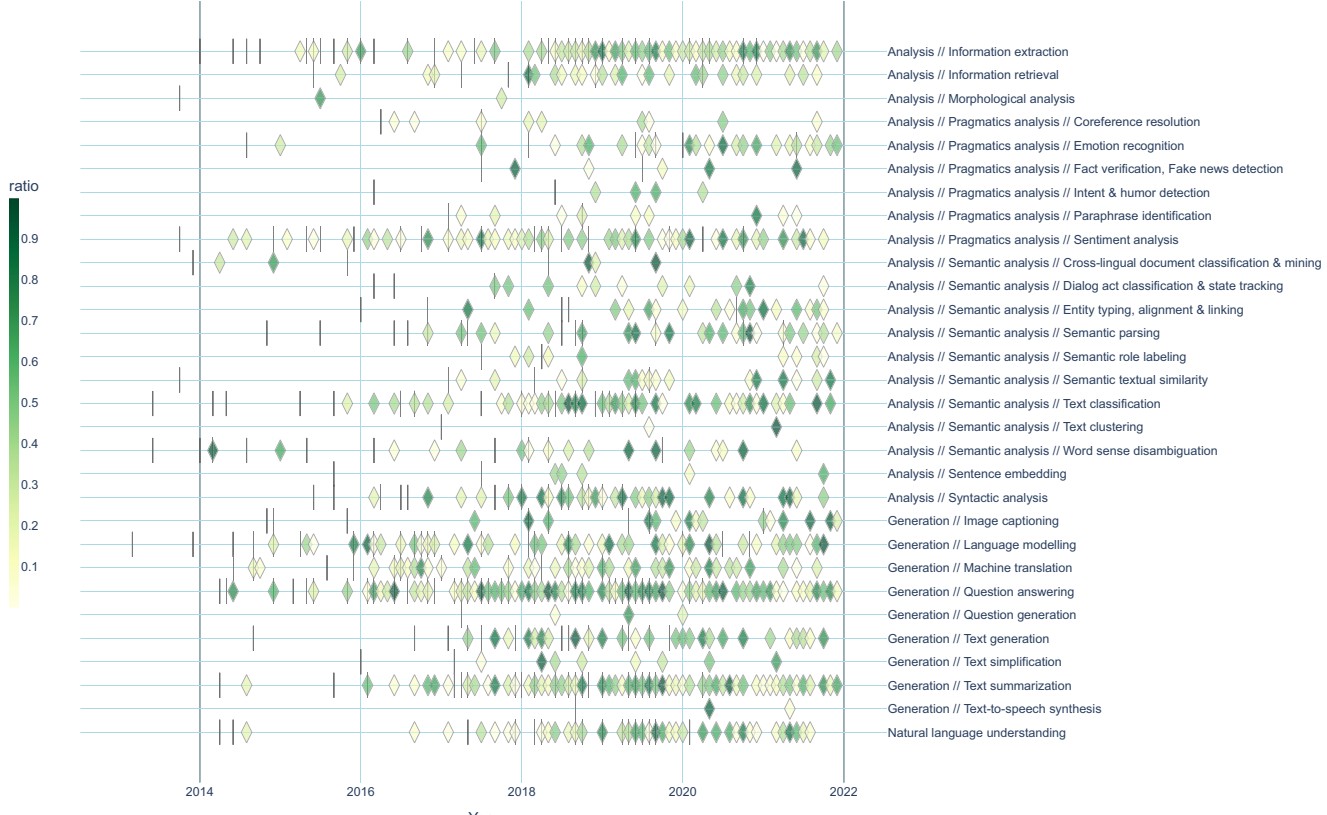

**Fig. 6 | Global SOTA improvement map for NLP.** Vertical dashes represent 'anchors', i.e., first results establishing a new benchmark. Diamond-shaped icons represent gains in a SOTA trajectory. Icon colors represent the relative improvements in SOTA for a specific benchmark as described in Fig. 5. Each task may contain data on multiple benchmarks, which are superimposed. Benchmarks containing fewer than three results at different time points and AI tasks that would contain only a single icon are not displayed. Detailed information for each data point (such as benchmark names) can be viewed in the interactive online versions of these figures at https://openbiolink.github.io/ITOExplorer/. A similar plot for computer vision, as well as plots aggregated by high-level task classes, are available in the supplementary figures and interactive online material.

Papers With Code (annotation data is available in Supplementary Data 8).

We found that even though dataset papers were highly cited, only a small fraction of citing papers reported results on the associated benchmarks, and an even smaller fraction (14 of 365, i.e., 3.84%) reported novel SOTA results. This implies that an estimated 0.0384 * 7595 = 291.32 papers in the combined corpus are expected to contain SOTA results. (Note: values here are shown rounded to two decimal places for ease of reading, but calculations were done with more precise numbers. Precise calculations are included in Supplementary Data 7.)

Meanwhile, Papers With Code contained SOTA results from 95 papers, i.e., 95/7595 = 1.23% of the combined corpus.

Taken together, 95/291.31 = 32.61% of papers containing SOTA results in the combined corpus were captured by Papers With Code, i.e., a coverage of approximately 1/3 of all SOTA results. While this indicates significant remaining potential for further increasing the coverage of Papers With Code, we deem this coverage sufficient to allow for meaningful aggregated analyses.

**Dataset attributes associated with popularity**

The finding that a large fraction of research activity was focussed on a comparatively small number of benchmark datasets and that many datasets failed to find adoption raises the question: which attributes differentiate highly popular from unpopular benchmark datasets? Gaining an understanding of these differences may guide creators of future benchmark datasets in prioritizing their efforts.

We conducted an exploratory analysis of some such potentially differentiating attributes. We selected all benchmark datasets used to benchmark NLP or computer vision tasks and which had first reported results in the Papers With Code dataset in 2018. We ranked selected datasets in two separate lists for NLP and computer vision by the number of unique papers that reported benchmark results on each dataset, i.e., a list ranked by the follow-up utilization of datasets for benchmarking.

We created two samples of top 10 and bottom 10 datasets (i.e., datasets with highest/least follow-up utilization for benchmarking) for NLP and computer vision, respectively (see "Methods" section for details on the sampling methodology). We combined the top and bottom lists of computer vision and NLP, resulting in a top list and a bottom list with 20 datasets each, yielding a total of $N = 40$ annotated datasets.

The majority of included datasets ($n = 36$; 90%) were associated with peer-reviewed publications. 33 datasets (82.5%) were associated with a paper with a first or last author with a public/academic affiliation, and 11 (27.5%) of datasets were associated with a paper with a first or last author with a private/industrial affiliation.

We investigated seven attributes for their correlation with top or bottom popularity status, based on the following driving hypotheses:

1. **Number of task types**, i.e., the number of different AI tasks that were evaluated by building on a specific dataset. This can include

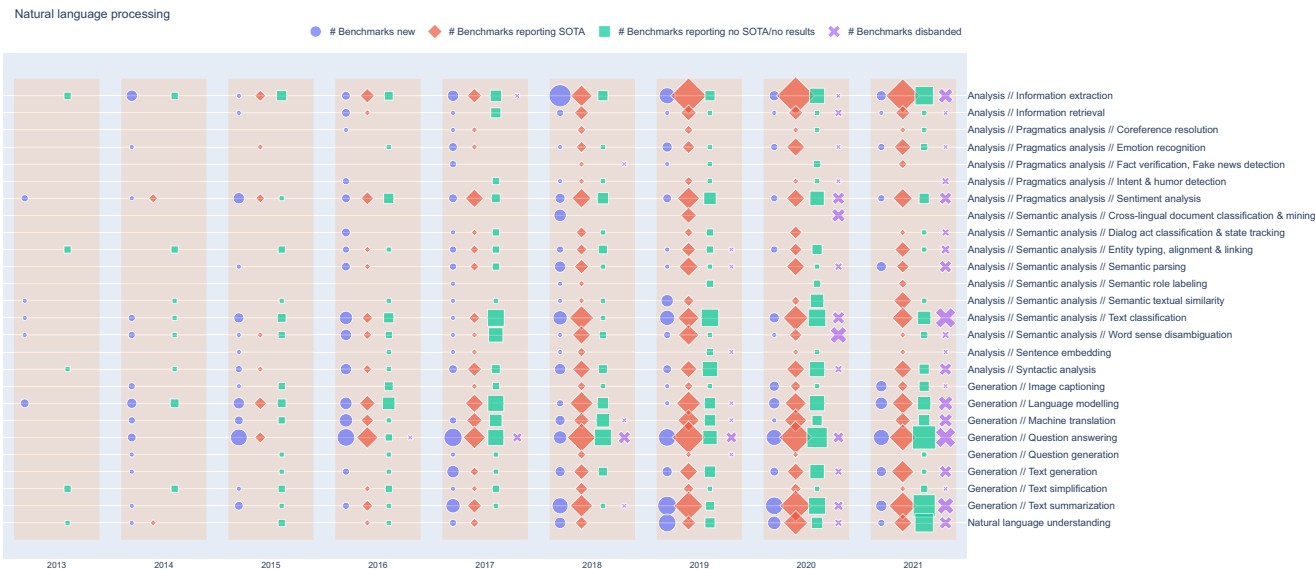

**Fig. 7 | AI benchmark lifecycle map for NLP.** Benchmarks with fewer than three reported results in at least one metric and tasks containing only a single benchmark are omitted. A similar plot for computer vision is available in the supplementary figures and interactive online material. SOTA: State-of-the-art.

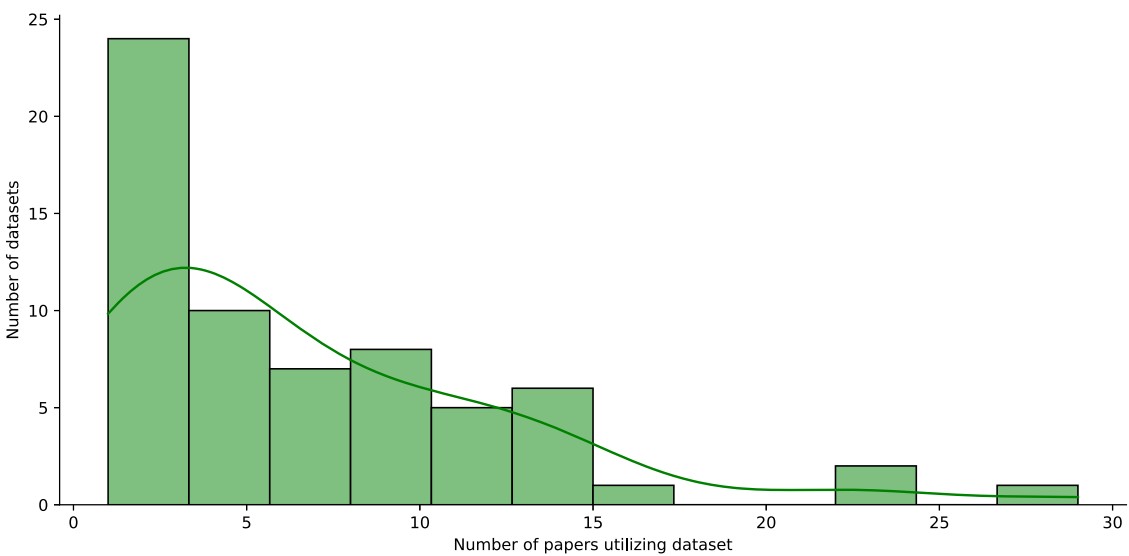

**Fig. 8 | Distribution of NLP dataset popularity.** Dataset popularity is measured by the number of scientific papers utilizing each dataset for which first results are reported in 2018.

tasks that were not originally envisioned during benchmark creation (dataset repurposing). *Hypothesis:* Top datasets have a higher number of task types. *Rationale:* Datasets that are flexible enough to be employed for a larger number of tasks types will see greater utilization.

2. **Number of sub-benchmarks.** Some benchmark datasets are made up of a predefined set of independent benchmark datasets. For example, the SuperGLUE NLP benchmark is made up of eight sub-benchmarks covering five different task types. *Hypothesis:* Top datasets have a higher number of sub-benchmarks. *Rationale:* Datasets that provide multiple benchmarks are more attractive because they cover a wider range of capabilities and are less prone to quick saturation.

3. **Dedicated leaderboard,** i.e., the dataset publishers advertised a dedicated, publicly available leaderboard. *Hypothesis:* Top datasets are more likely to have a dedicated leaderboard. *Rationale:* Providing a public leaderboard incentivizes benchmark use;

leaderboard provision is also a proxy for more elaborate and user-friendly setup of a benchmarking datasets.

4. **Proposed as part of a competition,** e.g., Kaggle, a workshop competition etc. *Hypothesis:* Top datasets are more likely to have been proposed as part of a competition. *Rationale:* Competitions lead to an initial burst of interest in the research community; this might also lead to larger follow-up interest of the community after the competition has ended.

5. **Top conference or journal,** i.e., the dataset paper was published in a top conference or journal (top status is defined through lists in Supplementary Data 6). *Hypothesis:* Top datasets are more likely to have been published in top conferences or journals. *Rationale:* Publication in top conferences or journals is a marker for higher quality datasets; datasets published in these venues are reaching a broader and more active audience.

6. **Number of institutions,** i.e., number of different institutions represented by co-authors of a dataset paper). *Hypothesis:* Top

**Table 2 | Comparison of datasets in the top vs. bottom popularity lists**

| | Top datasets (n = 20) | Bottom datasets (n = 20) | p |
|---|---|---|---|
| Number of associated publications | **14** (9–22) | **2** (1–3) | 0.000 |
| Number of task types | **2** (1–5) | **1** (1–2) | 0.007 |
| Number of sub-benchmarks | **2** (1–8) | **1** (1–1) | 0.015 |
| Dedicated leaderboard | **35%** | **0%** | 0.002 |
| Proposed as part of competition | **10%** | **15%** | 0.322 |
| Number of institutions | **2** (1–8) | **1** (1–6) | 0.310 |
| First/last author affiliated with top company/ university | **50%** | **20%** | 0.024 |

Datasets were sampled from NLP and computer vision datasets with first reported results in Papers With Code in 2018. Popularity was assessed by the number of publications that report benchmark results based on a dataset and are captured in the Papers With Code repository. Numeric attributes are reported as fractions or median values printed in bold. For median values, minimum and maximum values are shown in brackets.

datasets have a higher number of institutions. *Rationale:* The creation of good datasets requires broad collaboration; having a broader set of participants increases visibility in the community.

7. **Top company or university**, i.e., first or last authors are affiliated with a top-tier university or a company that is a key player in the AI domain. *Hypothesis:* Top datasets are more likely to have the first or last author affiliated with a top company or university. *Rationale:* Researchers at such institutions design datasets that are more broadly relevant and of higher utility; association with top institutions might increase interest of other researchers, positively impacting adoption.

A comparison of datasets in the top vs. bottom popularity lists is shown in Table 2. We found that datasets in the top popularity list were versatile (had greater number of task types), were published alongside a dedicated leaderboard, and had a larger number of sub-benchmarks (which was particularly the case for NLP datasets). Involvement of first/ last authors from top institutions was associated with greater popularity. Proposing benchmark datasets as part of a competition was not associated with greater popularity, as was the involvement of a greater number of institutions.

## Discussion

First, we found that a significant fraction of benchmarks quickly trends towards stagnation/saturation, and that this effect was especially marked in the recent past. One approach towards extending the useful lifetime of benchmarks could be an increased focus on benchmarks covering a larger number of sub-benchmarks covering different data distributions and task types. An extreme example is the recently released BIG-Bench benchmark (Srivastava et al. 2022), which contains >200 crowdsourced sub-benchmarks. Another approach could be the creation of 'living benchmarks' that are updated over time to prevent overfitting and benchmark saturation[10]. This could be achieved by creating tight loops of humans and AI systems working together on benchmark creation and evaluation[10,11]. However, it remains to be seen if this approach is practical enough to be adopted widely.

Second, we found that dynamics of performance gains on specific AI tasks usually do not follow clearly identifiable patterns. This indicates that progress in AI as captured by improvements in SOTA benchmark results remains rather unpredictable and prone to unexpected bursts of progress and phases of saturation/stagnation. This is likely caused both by the complexities and limitations of current benchmarking practices, as well as actual sudden bursts in AI capabilities. Deep learning models are flexible enough that 'cross-pollination' between developments in very different tasks and application domains is possible. For example, during the burst of progress in computer vision, developments from computer vision were transferred to NLP (e.g., convolutional neural networks applied to text classification[12]) while developments were transferred in the other direction during the more recent burst of progress in NLP (e.g., vision transformers[13]).

Third, we found that a significant fraction of benchmarks in our dataset was only utilized by a small number of independent studies at different time points. While this might be amplified by the incompleteness of the dataset, it does point towards an actual failure of many benchmarks to find widespread adoption. This resonates with recent findings that benchmarking efforts tend to be dominated by datasets created by a few high-profile institutions[9]. On the one hand, this raises concerns about potential bias and insufficient representativeness of benchmarks. On the other hand, recent criticism of the validity of many benchmarks for capturing real-world performance of AI systems[3] suggest that the development of fewer, but more quality-assured benchmarks covering multiple AI capabilities might be desirable[14].

Are current benchmarks covering all important AI tasks or are there fundamental gaps? This question cannot satisfactorily be answered by looking at benchmarking activity alone, and requires an in-depth analysis of the requirements of important AI application domains. As an example, we recently conducted a study in which we compared the explicitly stated needs for AI automation of clinical practitioners with the landscape of available clinical benchmark datasets[15]. We found that benchmark datasets failed to capture the needs of this user group, and that benchmarks for tasks that were most urgently required (such as assistance with documentation and administrative workflows) were missing completely. It is very plausible that similar misalignments between AI benchmarking practices and actual priorities for AI automation also exist in other domains.

Based on our findings and considerations, we can formulate some recommendations for creating benchmarks that are useful and utilized. Benchmarks should ideally be versatile, so that they can be used and re-purposed for a multitude of tasks. They should, if feasible, contain several sub-benchmarks covering different task types to decrease overfitting to narrowly defined tasks and to extend the lifespan of the benchmark by avoiding saturation from highly specialized models[1,11]. Benchmark creators should establish a leaderboard; we recommend establishing the benchmark directly in the Papers With Code platform and advertising that follow-up work should report results there. However, if feasible, benchmark performance should not be aggregated into a single metric but should be reported as a collection of metrics measuring different aspects of performance to avoid over-optimizing for specific metrics[16]. Benchmark creators should invest significant effort into orienting on most pressing use-cases and try to achieve high ecological validity, rather than merely orienting on easy access to existing data sources.

Our analyses have some important limitations. The curation of benchmark results across the entirety of AI research is highly labor-intensive. We therefore base our analysis on data from Papers With Code which—while being the most comprehensive source of benchmark data to date—still cannot provide a fully complete and unbiased representation of all benchmarking efforts in AI. A recent analysis concluded that while Papers With Code displays some bias towards recent work, its coverage of the literature is good and omitted works are mostly low-impact (as judged by citation count)[9]. We further investigated the completeness of Papers With Code in our study, and found that it covered approximately 1/3 of published SOTA results. While we deem this level of data completeness sufficient for the aggregated analyses of benchmarking dynamics as part of the present study, it underlines that significant improvements can still be made.

We therefore suggest that publishing research results to the Papers With Code repository should be further incentivized (e.g., through editorial guidelines of conferences and journals).

Some of our analyses put emphasis on the 'SOTA front', i.e., benchmark results that push the curve of SOTA results, while putting less emphasis on the dynamics of results below this SOTA curve. There are several good arguments that non-SOTA results can also provide valuable contributions and should receive more attention. For example, models that do not improve on SOTA performance metrics might have other benefits, such as better interpretability, lower resource need, lower bias or higher task versatility[1]. Nonetheless, research practice and scientific reward systems (such as paper acceptance) remain heavily influenced by progress on SOTA performance metrics, making their dynamics and potential shortcomings important subjects of investigation.

The creation of a global view on SOTA performance progress proved to be fraught with many difficulties. Performance results are reported for an enormous variety of tasks and benchmarks through an enormous variety of performance metrics that are often of questionable quality[16,17]. While we addressed some of these issues through a large-scale curation effort[18], the fundamental difficulty of interpreting the actual practical relevance, generalizability and potential impact of AI benchmark results remains. The analyses conducted in this work are therefore primarily geared towards furthering our understanding of the practice of AI benchmarking, rather than AI capability gain in itself. To better understand the real-world implications of specific benchmark results, more work needs to be done to map specific benchmark performance results to expected real-word impact—a currently very undeveloped field of investigation that should be the focus of future research.

## Methods

We extracted data from Papers With Code and conducted extensive manual curation to create a resource termed the Intelligence Task Ontology and Knowledge Graph (ITO)[18]. ITO broadly covers the results of different AI models applied against different benchmarks representative of different AI tasks in a coherent data model, and served as the basis of the analyses conducted in this study. AI tasks are grouped under major top-level classes in a rich task hierarchy. We queried benchmark results and task classifications and extracted SOTA values per metric spanning more than a decade (2008–mid 2021).

Metrics captured in ITO can have positive or negative polarities, i.e., they reflect performance improvement either as an increase (positive polarity, e.g., "Accuracy") or a decrease (negative polarity, e.g., "Error") in value. As we intended to depict an aggregated trajectory for both positive and negative polarities, we needed to detect and normalize the polarity of a large number of metrics. We identified metrics polarities through leaderboard analysis and manual curation, and inverted results with negative polarities prior to further analysis.

During curation, we found 662 metrics in ITO that were used with an unambiguous polarity, whereas 87 were utilized in apparently conflicting ways (reported with negative and positive polarity in different benchmarks because of data quality issues). We resolved this ambiguity through manual curation. A list with the 85 manually curated polarity assignments and the final list with the complete metrics can be found in Supplementary Data 1 and 2, respectively.

For the creation of the Intelligence Task Ontology and Knowledge Graph / ITO, we utilized the official data dump of the Papers With Code repository (data export date 2022/07/30) and devised Python scripts to convert the data into the Web Ontology Language (OWL) format[19]. We conducted extensive manual curation of the task class hierarchy and the performance metric property hierarchy in the collaborative ontology editing environment WebProtége[20] (version 4.0.2).

Data from ITO was loaded into the high-performance graph database BlazeGraph (version 2.1.6, blazegraph.com) and queried using the graph query language SPARQL[21]. For data manipulation, we used Pandas (version 1.2.4) for manipulating large data frames and Numpy (version 1.20.3) for numeric calculations (such as average). Plotly (version 4.14.3) was used for data visualization in order to create trajectory plots and state of the art curves on Jupyter™ Notebooks (version 6.4.0) running Python (version 3.9.5). We created an interactive Global Map of Artificial Intelligence Benchmark Dynamics using the graphical library Plotly.express (version 5.10, plotly.com/python/plotly-express) in dedicated Jupyter notebooks using Python (see Code availability)

For the creation of SOMs we used the Python library MiniSom (version 2.3.0, github.com/JustGlowing/minisom). We used Tunc's implementation (www.kaggle.com/izzettunc/introduction-to-time-series-clustering) to analyze our trajectories. The SOM parameters for the time series clustering were sigma = 0.3, learning rate = 0.1, random weight initialization and 50,000 iterations (see Code availability). For retrieving the top-k most similar trajectories to predefined functions we used tslearn (https://github.com/tslearn-team/tslearn/). Trajectories were first resampled to daily frequency, while missing values are filled with previous values (forward fill). Values were normalized to the range [0,1] using min-max normalization. Finally, the trajectory was resampled again into the interval $x \, \mathbb{N} \mid 0 \times 1200$.

For the creation of top and bottom dataset popularity lists, we split the ranking into two groups of most and least utilized datasets, such that the group of least utilized datasets have roughly the same utilization (i.e., number of papers utilizing the benchmark) as the group of most utilized datasets. Consider $P$ and $D$ as two lists, such that $P_i$ is the amount of unique papers utilizing the dataset $D_i$, and $P$ is sorted in descending order. We split the list of datasets $D$ at index $k$, such that $sum(P[0:k]) \approx sum(P[k:n])$ into two groups $D_- = D[0:k]$ and $D_+ = D[k:n]$. As $len(D_-) \gg len(D_+)$, we randomly subsampled elements from $D_-$ such that $len(D_-) = len(D_+)$. For both NLP and computer vision, we each created two lists of top-10 and bottom-10 datasets by using only the the top 10 and bottom 10 datasets from $D_+$ and $D_-$ respectively.

### Statistics
For comparing datasets in the top vs. bottom popularity sets (Table 2, Supplementary Data 6) we conducted unpaired, one-sided, heteroscedastic $t$-tests.

### Reporting summary
Further information on research design is available in the Nature Portfolio Reporting Summary linked to this article.

## Data availability
Source data are provided with this paper. Benchmark data was downloaded from the Papers With Code repository available at https://github.com/paperswithcode/paperswithcode-data (export date 2022/07/30). The curated knowledge graph underlying our analyses was deposited In the Zenodo database under accession code https://doi.org/10.5281/zenodo.7097305a [22]. The supplementary data generated in this study have been deposited in the Zenodo Database under accession code https://doi.org/10.5281/zenodo.7110147 [23].

## Code availability
The code for reproducing the results and generating interactive graphs is available from GitHub at: https://github.com/OpenBioLink/ITO/tree/master/notebooks/.

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

## Acknowledgements

This work was supported by netidee grant number 5158 (A.B., S.O.), European Community's Horizon 2020 Programme grant number 668353 (U-PGx) (K.B.), the EPSRC Centre for Doctoral Training in Autonomous Intelligent Machines and Systems (EP/S024050/1) (J.B.) and by Cancer Research UK (J.B.). We thank Robert Stojnic (Papers With Code) for his feedback.

## Author contributions

S.O. performed major analyses ex post manuscript review, produced figures, tables, supplementary material and codes available, wrote parts of the manuscript and approved the manuscript. A.B.-S. designed analyses, performed major analyses a priori manuscript review, produced figures, tables, supplementary material and codes available, wrote parts of the manuscript and approved the manuscript. K.B. performed data curation, produced figures, wrote parts of the manuscript and approved the manuscript. J.B. gave feedback on the analyses and the manuscript and approved the manuscript. M.S. designed the study, supervised all stages of the project, performed analyses, produced figures, tables, wrote major parts of the manuscript and approved the manuscript.

## Competing interests

The authors declare no competing interests.
