## [Peer Review File · Nature Communications]

Mapping global dynamics of benchmark creation and saturation in artificial intelligenceREVIEWER COMMENTS

Reviewer #1 (Remarks to the Author):

The manuscript examines the landscape of AI benchmarking datasets, with regards to their respective creation, usage/lifespan, and adoption by the research community. Specifically, they extracted and curated data from Papers with Code (PWC), an online repository of AI datasets, and investigated various aspects of AI datasets in two research areas: natural language processing (NLP) and computer vision (CV). For their analysis, they utilized clustering and developed their own method for measuring relative performance gains. Many of their findings are seemingly consistent with those previously reported. I have the following concerns/suggestions for the authors to consider:

1. The Table 1 results in this work show that AI datasets generally show different patterns in terms of community adoption. This is not surprising. What is the more interesting would be to understand the reasons behind the distinct usage; tips for developing a popular dataset; or lessons for avoiding abandonment.
2. The SOTA analysis suggests that “most benchmarks quickly trends towards the ceilings of their dynamic ranges, and that stagnation followed by burst patterns have been rare.” But it is not clear what kinds of AI tasks in NLP/CV fall into each category, even with the additional results in Figure 5.
3. Because sometimes there are multiple “anchors” in Figure 5, it is difficult to see which arrow corresponds to which benchmark. It is extremely confusing for “busy” tasks such as question answering. Moreover, why would an arrow appear before an anchor in some cases? This is not explained or is it an error?
4. For grouping and organizing similar NLP tasks shown in Figure 5, what is the difference between natural language generation: text summarization vs. the two text summarization tasks at the bottom. Also, there is Chinese NER but what about other language tasks such as Spanish, which are clearly present in PWC?
5. While both NLP and CV are used as case studies, only the former is described extensively in the main article – Note that AI and the recent Deep Learning algorithms have shown great success and potentials in CV research. There are probably more impressive breakthroughs in CV research than those in NLP. Is there a reason for this choice? Please justify.
6. Following #5, I think the contrast between AI in NLP and CV would be interesting to report here. In fact, many DL methods (e.g. GAN) were first developed for CV tasks and later adapted for NLP.
7. Given the nature of this work, it would be great if gaps in existing AI datasets could be identified. For instance, are there (sub-)domains or problems in NLP/CV research need special attention and/or more datasets? And what kind of datasets are of particular interest for future AI research? e.g. some known issues about AI model interpretability and generalizability. I feel this work would greatly benefit such analysis.
8. Note that a recent trend is to evaluate AI models on a collection of multiple datasets (e.g. the widely used GLUE in NLP research). does this trend have any implications for the analysis in this work?

Minor:

most benchmarks quickly trends should be trend

Reviewer #2 (Remarks to the Author):

This paper surveys the benchmarks and results on those benchmarks over time within the papers with code collection. It classifies benchmarks as being saturated, enjoying steady progress, or seeing bursty progress. It provides visualization of the rate of progress on different benchmarks over time.

Benchmarks are the critical "loss function" of the ML research community. Analysis of the rate of progress on benchmarks has significant implications for the field as a whole and is worthy of study. It complements the work of Koch (cited), but focused on benchmark progress more than benchmark usage. The author's analysis is sound and easy to follow. Their conclusions that benchmarks are saturating more quickly and could benefit from broader support have been presented elsewhere (e.g. the Dynabench paper) but those conclusions are important and the authors establish broader supporting evidence.

My primary criticism of the work is that the authors primarily provide statistical support conclusions already raised, and forego diving more deeply into characterization of the benchmarks they chart. I would love to see hierarchical taxonomy of benchmarks leading to analysis at a group level, especially welcome with the plethora of NLI benchmarks. Further, analysis of characteristics such as dataset size, source etc. would be useful. From this, I believe the authors could draw more interesting conclusions about the state of the benchmarking ecosystem and its direction of future evolution.

In general, I believe the work is in keeping with the standards of the field and sufficient to be a useful contribution, but wish it went deeper in its analysis especially given overlap with existing work. This is a rich and important area for exploration.

Reviewer #3 (Remarks to the Author):

The paper shows several findings about the artificial intelligence benchmarks in the recent past. First, the majority of them quickly trend to near- or total saturation/stagnation. Second, a "significant fraction of these benchmarks" were only utilized by a small number of independent studies. Third, the paper claims that the dynamics of performance gains on benchmarks mostly do not follow clearly identifiable patterns. These findings/conclusions were based on data extracted from Paper with Code (paperswithcode.com), and updated from authors' previous publication termed intelligence task ontology and knowledge (ITO).

These findings are demonstrated in the paper based on some statistical analysis, while many researchers in the artificial intelligence field may already feel the same way. For example, I suspect most of the researchers who seriously work on image classification would know ImageNet ("Imagenet: A large-scale hierarchical image database", CVPR 2009) is about to be saturated. I don't think it is hard to obtain these findings, yet I think it is good to organize the statistical analysis over hundreds of recent benchmarks.

I have two suggestions for the authors. First, I wish the authors could give some more suggestions about how to construct more meaningful benchmarks based on the findings in the paper. Second, if some benchmark is mentioned specifically in the paper (i.e. ImageNet), appropriate citation should be added.

We thank the reviewers for their suggestions that helped us improve our manuscript. We significantly extended our analyses and revised the manuscript accordingly.

Major additions and changes are highlighted through green bars on the left side of the text in the main manuscript; specific locations of major changes are identified with markers that we reference here, e.g.:

[change_1]
Changed text

Reviewer #1 (Remarks to the Author):

“The manuscript examines the landscape of AI benchmarking datasets, with regards to their respective creation, usage/lifespan, and adoption by the research community. Specifically, they extracted and curated data from Papers with Code (PWC), an online repository of AI datasets, and investigated various aspects of AI datasets in two research areas: natural language processing (NLP) and computer vision (CV). For their analysis, they utilized clustering and developed their own method for measuring relative performance gains. Many of their findings are seemingly consistent with those previously reported. I have the following concerns/suggestions for the authors to consider:

1. The Table 1 results in this work show that AI datasets generally show different patterns in terms of community adoption. This is not surprising. What is the more interesting would be to understand the reasons behind the distinct usage; tips for developing a popular dataset; or lessons for avoiding abandonment.”

→ **Thanks for pointing out this practical consideration. We added an analysis in which we manually annotated a set of benchmark datasets in order to find attributes associated with benchmark dataset utilization / popularity. We also added suggestions for dataset development to the discussion section.**

Location of changes: [change_5] and [change_6_results] in the results section, [change_6_methods] in the methods section and [change_7] in the discussion section of the main manuscript; addition of Suppl. Table 11 in the Supplementary Data file.

“2. The SOTA analysis suggests that “most benchmarks quickly trends towards the ceilings of their dynamic ranges, and that stagnation followed by burst patterns have been rare.” But it is not clear what kinds of AI tasks in NLP/CV fall into each category, even with the additional results in Figure 5.”

→ We extended the analysis to capture the entirety of benchmark results in our dataset (the previous analysis focused benchmark results reported as ‘accuracy’). We generated exemplary top-10 lists of those benchmarks with SOTA curves most prototypical for each dynamics pattern (i.e. continuous growth, saturation/stagnation, stagnation followed by burst), as well as the tasks associated with these benchmarks. However, global analyses of the associations between benchmark tasks and dynamic patterns did not reveal practically significant associations.

Location of changes: [change_1] in the results section of the main manuscript.

“3. Because sometimes there are multiple “anchors” in Figure 5, it is difficult to see which arrow corresponds to which benchmark. It is extremely confusing for “busy” tasks such as question answering.

→ The primary aim of these figures is to provide a condensed view of the benchmarking activity for different tasks, across individual benchmarks. Indeed, detailed per-benchmark information is not conveyed when these graphs are viewed in print. However, this information can be explored through the interactive versions of these graphs on the associated webpage (<https://openbiolink.github.io/ITOExplorer/>) and Jupyter notebooks (Code 2, Code Availability).

We added references to the interactive webpage in the figure captions, encouraging readers to explore the interactive visualizations for more in-depth analysis.

“Moreover, why would an arrow appear before an anchor in some cases? This is not explained or is it an error?”

→ Thanks for pointing out this potential source of confusion. The date of the benchmark result in the figure was signified through the right-side tip of the arrow rather than the left side of the arrow, which we admit was counterintuitive and was the reason for confusing cases like this.

We revised the figure, which now uses different icons without this issue.

Location of changes: [change_2] and [change_3] in the results section of the main manuscript.

“4. For grouping and organizing similar NLP tasks shown in Figure 5, what is the difference between natural language generation: text summarization vs. the two text summarization tasks at the bottom.”

→ Both 'Document summarization' and 'Abstractive summarization' are subclasses of 'Text summarization', which in itself is a subclass of 'Natural language generation' in the ontology. Benchmark results can be assigned either to the more general parent classes or to the more specific subclasses. In this case, some of the benchmark results were categorized in the most specific subclasses, which led to this rendering in Figure 5.

“Also, there is Chinese NER but what about other language tasks such as Spanish, which are clearly present in PWC?”

→ We indeed have Spanish NER tasks in the dataset, however they are not displayed in the figure because of the filtering criteria that were applied when the data was extracted from PWC and there was not sufficient activity recorded for these tasks (e.g., we only show benchmarks with a minimum amount of activity over time; see Table 1 for reference).

“5. While both NLP and CV are used as case studies, only the former is described extensively in the main article – Note that AI and the recent Deep Learning algorithms have shown great success and potentials in CV research. There are probably more impressive breakthroughs in CV research than those in NLP. Is there a reason for this choice? Please justify.”

→ We added further discussion of findings for computer vision to the results section.

Location of changes: [change_4] in the results section of the main manuscript.

While breakthroughs in CV initiated the deep learning revolution, we think that NLP has caught up in terms of progress and impact in more recent years, with both domains undergoing tremendous progress and receiving widespread attention. We chose to give preference to plots for the NLP domain in the main manuscript for the pragmatic reason that the respective plots for computer vision would be too large to be easily displayed within the main manuscript.

We hope that readers will be satisfied with the interactive online versions of these graphs, and have added more pointers to these online graphs to the main manuscript. Furthermore, the figures for CV are contained in the supplementary figures file.

“6. Following #5, I think the contrast between AI in NLP and CV would be interesting to report here. In fact, many DL methods (e.g. GAN) were first developed for CV tasks and later adapted for NLP.”

→ We added a brief discussion of the dynamics between NLP and CV to the discussion section. We argue that historically, ideas from CV were flowing into NLP, while after the more recent breakthroughs in NLP, ideas from NLP are flowing into CV. We expect the

two domains to become increasingly integrated as models become more versatile and multimodal. We hope that this discussion is sufficient to sensitize readers to such potential underlying dynamics; it would unfortunately be difficult to quantify such an interplay of model architectures.

Location of changes: [change_7] in the discussion section.

“7. Given the nature of this work, it would be great if gaps in existing AI datasets could be identified. For instance, are there (sub-)domains or problems in NLP/CV research need special attention and/or more datasets? And what kind of datasets are of particular interest for future AI research? e.g. some known issues about AI model interpretability and generalizability. I feel this work would greatly benefit such analysis.”

→ **We added a discussion of potential gaps in the dataset and benchmark landscape and how they can be identified. We cite and discuss a related, more targeted study of our group in which we identify gaps in benchmark datasets in AI for medical routine. We argue that such gaps can be substantial, and that their identification necessitates studies that actively assess needs through studying potential application domains and user needs.**

Location of changes: [change_7] in the discussion section.

“8. Note that a recent trend is to evaluate AI models on a collection of multiple datasets (e.g. the widely used GLUE in NLP research). does this trend have any implications for the analysis in this work?”

→ **Our benchmark dynamics analyses and graphs (such as Fig. 5 and Fig. 6) operate on the level of tasks and task types. The tasks/datasets covered by the ‘sub-benchmarks’ in e.g. GLUE are represented and visualized independently in our benchmark dynamics maps, and these visualizations are therefore not directly impacted by such benchmarks.**

We strongly agree that investigating the impact of these types of benchmarks would be of interest. In this revision, we added an exploratory analysis of the impact of benchmarks having multiple ‘sub-benchmarks’ on benchmark popularity

Location of changes: [change_6_results], including Table 2.

“Minor:
most benchmarks quickly trends should be trend”

→ **We rewrote this sentence.**

Reviewer #2 (Remarks to the Author):

“This paper survey the benchmarks and results on those benchmarks over time within the papers with code collection. It classifies benchmarks as being saturated, enjoying steady progress, or seeing bursty progress. It provides visualization of the rate of progress on different benchmarks over time.

Benchmarks are the critical "loss function" of the ML research community. Analysis of the rate of progress on benchmarks has significant implications for the field as a whole and is worthy of study. It complements the work of Koch (cited), but focused on benchmark progress more than benchmark usage. The author's analysis is sound and easy to follow. Their conclusions that benchmarks are saturating more quickly and could benefit from broader support have been presented elsewhere (e.g. the Dynabench paper) but those conclusions are important and the authors establish broader supporting evidence.”

→ **We thank the reviewer for their encouraging feedback!**

“My primary criticism of the work is that the authors primarily provide statistical support conclusions already raised, and forego diving more deeply into characterization of the benchmarks they chart. I would love to see hierarchical taxonomy of benchmarks leading to analysis at a group level, especially welcome with the plethora of NLI benchmarks.

→ **We have created additional versions of the “global SOTA improvement maps” for NLP and computer vision, in which results are aggregated to top-level tasks. Unfortunately these graphs cannot be added to the main manuscript because of space restrictions; they were added to the supplementary materials as Suppl. Figure 4 and 5.**

Location of changes: Supplementary information file.

“Further, analysis of characteristics such as dataset size, source etc. would be useful. From this, I believe the authors could draw more interesting conclusions about the state of the benchmarking ecosystem and its direction of future evolution.”

→ **We added an analysis in which we manually annotated a set of benchmark datasets with such attributes in order to find attributes associated with benchmark dataset utilization / popularity.**

Location of changes: [change_6_results] and [change_6_methods] in the result and methods sections of main manuscript; addition of Suppl. Table 11 in the Supplementary Data file.

Please note that some potentially interesting attributes are hard to analyze on a global scale, because necessary data are not available without a prohibitively large curation effort. We also found comparing some attributes (such as dataset size) across a wide variety of benchmark datasets to be problematic. The meaning of these attributes can strongly vary between different datasets and task types and providing fair comparisons quickly results in a lot of complexity. (Illustrative example of complexity arising even within a single task type: “One entity linking dataset contains 2000 linked entities in 500 sentences, another entity linking dataset contains 1000 linked entities in 1000 sentences – which one should be considered larger?”)

“In general, I believe the work is in keeping with the standards of the field and sufficient to be a useful contribution, but wish it went deeper in its analysis especially given overlap with existing work. This is a rich and important area for exploration.”

Reviewer #3 (Remarks to the Author):

“The paper shows several findings about the artificial intelligence benchmarks in the recent past. First, the majority of them quickly trend to near- or total saturation/stagnation. Second, a “significant fraction of these benchmarks” were only utilized by a small number of independent studies. Third, the paper claims that the dynamics of performance gains on benchmarks mostly do not follow clearly identifiable patterns. These findings/conclusions were based on data extracted from Paper with Code (paperswithcode.com), and updated from authors’ previous publication termed intelligence task ontology and knowledge (ITO).

These findings are demonstrated in the paper based on some statistical analysis, while many researchers in the artificial intelligence field may already feel the same way. For example, I suspect most of the researchers who seriously work on image classification would know ImageNet (“Imagenet: A large-scale hierarchical image database”, CVPR 2009) is about to be saturated. I don’t think it is hard to obtain these findings, yet I think it is good to organize the statistical analysis over hundreds of recent benchmarks.

I have two suggestions for the authors. First, I wish the authors could give some more suggestions about how to construct more meaningful benchmarks based-on the findings in the paper.”

→ **We now provide such suggestions in the discussion section, based on analyses of the first version of the manuscript, the novel analyses in this revision of the manuscript as well as recent relevant literature.**

Location of changes: [change_7] in the discussion section of the main manuscript.

“Second, if some benchmark is mentioned specifically in the paper (i.e. ImageNet), appropriate citation should be added.”

→ **Thanks for pointing out this oversight; we added these citations.**

REVIEWER COMMENTS

Reviewer #1 (Remarks to the Author):

I thank the authors for responding to my previous comments. Please see below my remaining concerns.

Major:

1. In the revised part, the author acknowledged data incompleteness in PWC as an important limitation of this study. They further cited that a previous study stating that "its (PMC) coverage of the literature is good and omitted works are mostly low-impact." This is helpful information but is insufficient for this work because (a) the coverage assessment of the cited study is purely based on citation count, and it is a rough approximation of true coverage at best. More importantly, data completeness plays the most critical role in the findings reported in this work (e.g., Table 1 results). After manually examining a few benchmark datasets (in both NLP & CV), the data coverage issue in PWC is concerning.

For instance, the BC4CHEMD dataset has two listed models in PWC <https://paperswithcode.com/sota/named-entity-recognition-on-bc4chemd> in fact, this is one of the most studied NER datasets in biomedical NLP and has received wide attention with hundreds of cited works from the research community. Not only does it miss many previous studies, it also does not have the accurate listing of SOTA results (current best is already over 95% in this case).

This issue can also be found in CV datasets. For example, the Refuge challenge dataset (glaucoma segmentation) is indexed with two studies <https://paperswithcode.com/sota/optic-disc-segmentation-on-refuge-challenge> higher SOTA results can be quickly found from the studies cited this dataset in a straightforward google search. Unfortunately, according to the current data in PWD, studies such as these would be grouped erroneously in various tables and figures presented in the manuscript.

Given the discrepancy in the underlying data, there is a need to quantify this critical limitation and its implications in the results in various tables and figures, as it has a direct impact on almost all of the findings. Take the numbers in Table 1 for example, I suspect the true fraction of benchmarks and AI tasks with three or more results are higher than those currently reported. Similarly, Figure 4 shows that with current data, nearly half of the benchmarks do not report novel SOTA results. This number may also be under-estimated.

2. While it may be easier to view the Fig. 6 information interactively online, the figure itself is still prone to cause confusion and not very informative. I'd suggest (a) either simplifying it with a representative dataset per task and/or (b) moving the full figure to the supplementary materials.

3. It would be beneficial to include the 2021 data, as the results would be at least two years old when the manuscript is accepted for publication.

Minor:

Please define "benchmark active" in the Figure 4 caption.

Page 9: Fig.s should be Figs.

Reviewer #2 (Remarks to the Author):

I am satisfied with the changes. The additional data on the benchmarks and type decomposition is useful and well presented. Thank you to the authors!

We thank the reviewers for their suggestions that helped us improve our manuscript.

We significantly extended our analyses based on Feedback for Reviewer #1 and revised the manuscript accordingly.

Major additions and changes are highlighted through green bars on the left side of the text in the main manuscript.

Reviewer #1 (Remarks to the Author):

“I thank the authors for responding to my previous comments. Please see below my remaining concerns.

Major:

1. In the revised part, the author acknowledged data incompleteness in PWC as an important limitation of this study. They further cited that a previous study stating that “its (PMC) coverage of the literature is good and omitted works are mostly low-impact.” This is helpful information but is insufficient for this work because (a) the coverage assessment of the cited study is purely based on citation count, and it is a rough approximation of true coverage at best. More importantly, data completeness plays the most critical role in the findings reported in this work (e.g., Table 1 results). After manually examining a few benchmark datasets (in both NLP & CV), the data coverage issue in PWC is concerning.

For instance, the BC4CHEMD dataset has two listed models in PWC

<https://paperswithcode.com/sota/named-entity-recognition-on-bc4chemd> in fact, this is one of the most studied NER datasets in biomedical NLP and has received wide attention with hundreds of cited works from the research community. Not only does it miss many previous studies, it also does not have the accurate listing of SOTA results (current best is already over 95% in this case).

This issue can also be found in CV datasets. For example, the Refuge challenge dataset (glaucoma segmentation) is indexed with two studies

<https://paperswithcode.com/sota/optic-disc-segmentation-on-refuge-challenge> higher SOTA results can be quickly found from the studies cited this dataset in a straightforward google search. Unfortunately, according to the current data in PWD, studies such as these would be grouped erroneously in various tables and figures presented in the manuscript.

Given the discrepancy in the underlying data, there is a need to quantify this critical limitation and its implications in the results in various tables and figures, as it has a direct impact on almost all of the findings. Take the numbers in Table 1 for example, I suspect the true fraction of benchmarks and AI tasks with three or more results are higher than those currently reported. Similarly, Figure 4 shows that with current data, nearly half of the benchmarks do not report novel SOTA results. This number may also be under-estimated.”

→ We have conducted an extensive additional curation effort and analysis to quantify the completeness of the Papers With Code dataset. We conclude that based on our analysis, PWC covers $\frac{1}{3}$ of the SOTA results in the total literature. We deem this coverage sufficient for the aggregated dynamics analyses across entire task classes and AI research domains conducted in our study, but also agree with the reviewer that coverage of PWC would profit from further improvement.

Doing the manual curation for this additional analysis demonstrated to us how labor-intensive the large-scale curation of benchmark results is—and how large therefore the value of the existing PWC dataset really is. We enhanced the manuscript with a short discussion of these issues, and added a recommendation for further enhancing the PWC dataset in the future.

We added the following section to the results section:

>>>

Quantifying Papers With Code dataset completeness

While Papers With Code is the largest dataset of AI benchmark results by a wide margin, it cannot provide a full coverage of all existing AI benchmarks. We conducted a small-scale study to estimate the completeness of the Papers With Code dataset regarding SOTA result trajectories.

We randomly sampled 10 benchmark datasets from NLP and 10 benchmark datasets from computer vision in the dataset, resulting in a total of 20 randomly sampled datasets (listed in Supplementary Data 7). Querying Google Scholar, we found that the total size of the combined corpus of papers introducing the datasets and all their citing papers was 7595. Out of the citing papers, we randomly sampled 365 papers (sample size chosen to yield a margin of error of 5% in the analysis).

We inspected and annotated these 365 papers to determine whether each paper contained results on the benchmark of the cited dataset paper. If this was the case, we compared the reported result with the Papers With Code dataset to determine if the paper reported a result that was SOTA at the time and was not currently covered by Papers With Code (annotation data is available in Supplementary Data 8)

We found that even though dataset papers were highly cited, only a small fraction of citing papers reported results on the associated benchmarks, and an even smaller fraction (14 of 365, i.e. 3.83%) reported novel SOTA results. This implies that an estimated $0.38 * 7595 = 291.32$ papers in the combined corpus are expected to contain SOTA results.

Meanwhile, Papers With Code contained SOTA results from 95 papers, i.e. $95 / 7575 = 1.23\%$ of the combined corpus.

Taken together, $95 / 291.31 = 32.61\%$ of papers containing SOTA results in the combined corpus were captured by Papers With Code, i.e. a coverage of approximately $\frac{1}{3}$ of all SOTA results. While this

indicates significant remaining potential for further increasing the coverage of Papers With Code, we deem this coverage sufficient to allow for meaningful aggregated analyses.

<<<

We also extended the associated discussion in the discussion section of the manuscript with the following text:

>>>

Our analyses have some important limitations. The curation of benchmark results across the entirety of AI research is highly labor-intensive. We therefore base our analysis on data from Papers With Code which—while being the most comprehensive source of benchmark data to date—still cannot provide a fully complete and unbiased representation of all benchmarking efforts in AI. A recent analysis concluded that while Papers With Code displays some bias towards recent work, its coverage of the literature is good and omitted works are mostly low-impact (as judged by citation count) 12. We further investigated the completeness of Papers With Code in our study, and found that it covered approximately $\frac{1}{3}$ of published SOTA results. While we deem this level of data completeness sufficient for the aggregated analyses of benchmarking dynamics as part of the present study, it underlines that significant improvements can still be made. We therefore suggest that publishing research results to the Papers With Code repository should be further incentivized (e.g., through editorial guidelines of conferences and journals).

<<<

“2. While it may be easier to view the Fig. 6 information interactively online, the figure itself is still prone to cause confusion and not very informative. I’d suggest (a) either simplifying it with a representative dataset per task and/or (b) moving the full figure to the supplementary materials.”

→ We have significantly revised Fig. 6 to make it easier to parse. We condensed the grouping of tasks to make the figure more compact, changed the layout so that the hierarchy of task labels is easier to parse. We also changed the aggregation method for multiple benchmark results in the same month and for the same task. Previously, we calculated the mean, now we take the maximum value. This way, bursts in progress in some benchmarks are accentuated, making more of the dynamics visible.

Several readers of the preprint of the previous manuscript communicated their interest in this figure, so we think it is preferable to keep it in the main manuscript and to not further simplify it to cater to this subset of readers.

“3. It would be beneficial to include the 2021 data, as the results would be at least two years old when the manuscript is accepted for publication.”

→ **We agree that refreshing the data adds to the value of the manuscript when accepted for publication. We decided to invest a significant amount of effort into updating the analyses and paper with recent data; doing the update entailed additional curation of the AI tasks and their hierarchy (which had further evolved in PWC), revising the code for the analysis pipeline, revising the figures and updating several parts of text across the manuscript. We are pleased to report that this effort paid off and we can now present the extensive analyses of the paper based on a very recent export from PWC.**

“Minor:

Please define “benchmark active” in the Figure 4 caption.

Page 9: Fig.s should be Figs.”

→ **We have updated the manuscript accordingly.**

Reviewer #2 (Remarks to the Author):

I am satisfied with the changes. The additional data on the benchmarks and type decomposition is useful and well presented. Thank you to the authors!

→ **We thank the reviewer for their time and effort!**

REVIEWERS' COMMENTS

Reviewer #1 (Remarks to the Author):

I thank the authors for the additional data analysis, among several other changes and updates.

A few minor corrections should be made on page 15. $0.38*7595=291.32$ should be $0.038*7595 = 288.61$; also, $95/7575$ should be $95/7595$? finally $95/291.31$  $95/288$.

Response to reviewer comments

“Reviewer #1 (Remarks to the Author):

I thank the authors for the additional data analysis, among several other changes and updates.

A few minor corrections should be made on page 15. $0.38 \cdot 7595 = 291.32$ should be $0.038 \cdot 7595 = 288.61$; also, $95/7575$ should be $95/7595$? finally $95/291.31 \rightarrow 95/288.$ ”

→ **We corrected a typo (0.38 -> 0.038). We made the clarification that calculations were made with precise numbers but are shown with limited precision for space reasons better visible (it was previously in a footnote and is now part of the main text). We also added an explicit reference to the supplementary data, in which precise calculations can be found.**